# Conflict-climate-displacement: a cross-sectional ecological study determining the burden, risk and need for strategies for neglected tropical disease programmes in Africa

Louise A Kelly-Hope  ,[1] Emma Michèle Harding-Esch  ,[2] Johan Willems,[3] Fatima Ahmed,[4] Angelia M Sanders  [5]

[1]Institute of Infection, Veterinary and Ecological Sciences, University of Liverpool, Liverpool, UK
[2]Department of Clinical Research, London School of Hygiene & Tropical Medicine, London, UK
[3]CBM Christoffel-Blindenmission Christian Blind Mission e.V, Bensheim, Germany
[4]The MENTOR Initiative, Haywards Heath, UK
[5]Trachoma Control Program, The Carter Center, Atlanta, Georgia, USA

**Correspondence to**
Dr Louise A Kelly-Hope;
lkhope@liverpool.ac.uk

## ABSTRACT

**Objectives** Complex challenges such as political instability, climate change and population displacement are increasing threats to national disease control, elimination and eradication programmes. The objective of this study was to determine the burden and risk of conflict-related and climate-related internal displacements and the need for strategies for countries endemic with neglected tropical diseases (NTDs).

**Design, setting and outcome measures** A cross-sectional ecological study was conducted including countries that are endemic with at least one of five NTDs requiring preventive chemotherapy in the African region. For each country, the number of NTDs, population size and the number and rate per 100 000 of conflict-related and natural disaster-related internal displacements reported in 2021 were classified into high and low categories and used in unison to stratify and map the burden and risk.

**Results** This analysis identified 45 NTD-endemic countries; 8 countries were co-endemic with 4 or 5 diseases and had populations classified as 'high' totalling >619 million people. We found 32 endemic countries had data on internal displacements related to conflict and disasters (n=16), disasters only (n=15) or conflict only (n=1). Six countries had both high conflict-related and disaster-related internal displacement numbers totalling >10.8 million people, and five countries had combined high conflict-related and disaster-related internal displacement rates, ranging from 770.8 to 7088.1 per 100 000 population. Weather-related hazards were the main cause of natural disaster-related displacements, predominately floods.

**Conclusions** This paper presents a risk stratified approach to better understand the potential impact of these complex intersecting challenges. We advocate for a 'call to action' to encourage national and international stakeholders to further develop, implement and evaluate strategies to better assess NTD endemicity, and deliver interventions, in areas at risk of, or experiencing, conflict and climate disasters, in order to help meet the national targets.

## STRENGTHS AND LIMITATIONS OF THIS STUDY

⇒ This study is the first to document the number of conflict-related and natural disaster-related internal displacements in high neglected tropical disease (NTDs)-endemic countries in Africa.
⇒ The methodology employed is based on publicly available resources downloaded from the Expanded Special Project for Elimination of NTDs data portal and the Internal Displacement Monitoring Centre.
⇒ A new practical risk stratification approach is presented to help determine the intersecting challenges of NTDs and conflict-related and natural disaster-related internal displacements, and highlights each country's unique combination of risk or vulnerability.
⇒ This information can be used for advocacy purposes to develop and implement mitigation strategies to help direct resources and enable effective NTD control and elimination programmes.
⇒ Future work needs to consider a range of data sources and spatial-temporal patterns to help identify hotspots, and determine the long-term implications and strategies needed for NTD programme activities.

## INTRODUCTION

The neglected tropical disease (NTD) road map 2021–2030 sets out aspiring global targets to prevent, control, eliminate or eradicate 20 NTDs in >100 endemic countries.[1] The roadmap highlights how reaching these goals can contribute to attaining the Sustainable Development Goals,[2 3] and emphasises the need to accelerate programmatic action, intensify cross-cutting approaches and change operating models and culture to facilitate country ownership. Importantly, it raises the need to address complex challenges such as climate change, political instability and population movement and displacement, which threatens achieving

the road map targets, and are increasing and intrinsically interlinked but have received little attention to date.

Climate change can alter local ecology and disease dynamics.[4–6] It is expected to increase the intensity and frequency of extreme weather events and induce natural disasters such as floods and drought, which can restrict access to healthcare, limit water and food systems, cause outbreaks of climate-sensitive diseases and lead to forced migration and internal displacement as living areas become inhabitable and livelihoods are destroyed.[7 8] The African continent is extremely vulnerable to climate change.[9] However, there is a major knowledge gap associated with its impact on NTD transmission and programme activities across highly endemic countries.

Political instability can disrupt governance and escalate internal conflict and violence, and therefore restrict safe access to healthcare and potentially cause forced migration and internal displacement as people seek refuge. In Africa, the number of conflict events has increased dramatically in recent years[10 11] and impeded the effectiveness of many NTD programmes.[12–15] In response, practical approaches have been developed to address key issues such as safe and effective disease mapping and intervention strategies,[16–18] however more support to implement these at scale, and better understand the intersect with climate and complex humanitarian emergencies leading to displacement and how it affects vulnerable populations is urgently needed.[19]

External and internal displacement are driven by multiple causes and pose multiple risks for national programmes.[20] They can result in the introduction or recrudescence of infectious diseases, and the shortage of essential services such as healthcare, water and sanitation in areas with large unplanned population fluxes. These factors make it difficult for programmes to implement and assess the impact of interventions such as mass drug administration (MDA), which is being widely distributed as preventive chemotherapy for a number of NTDs.[1 21] The lack of specific protocols for distributing MDA to internally displaced populations and little understanding of the inhibiting factors that programmes face exacerbates the problem.

If the NTD road map 2021–2030 global targets are to be achieved, these challenges that NTD programmes face need to be at the forefront of national plans and stakeholder funding in the next decade. Currently, there are no formal health policies or guidelines on how national programmes may begin to document the burden, assess the impact and develop strategies to overcome these pervasive threats, nor (to the best of our knowledge), specific funding streams exist for operational research and policy development. This is of significant concern and needs prioritising given that these challenges will drastically hinder progress, especially in Africa where many large, highly NTD-endemic countries have additional programme challenges and are lagging behind key programmatic activities.[1 22]

The frequency of these displacement-related events in many NTD-endemic countries is escalating, highlighting the need for urgent attention. The Internal Displacement Monitoring Centre (IDMC) highlights that conflict, violence and natural disasters caused approximately 38 million (M) internal displacements worldwide in 2021 with 80% of conflict and violence triggered displacements in sub-Saharan Africa alone.[23] Weather-related natural disasters such as storms and floods accounted for the vast majority of the 23.7 M natural disaster (94%)-internal displacements worldwide, underlining the threat of climate change.

NTD programmes need to better understand the burden and the extent to which these are singular or multiple intersecting challenges (figure 1) to help to develop bespoke strategies and direct resources appropriately. This is important as some countries will experience more conflict than climate, others more climate than conflict or a complex combination of both conflict-climate.

We conducted a cross-sectional ecological study to better understand the burden of conflict-related, and natural disaster-related internal displacements in NTD-endemic countries, and the extent to which these are singular or multiple intersecting challenges, to provide a basis from which bespoke strategies could be developed and resource allocation could be appropriately directed. We focused on countries in the WHO African region that are endemic for at least one of the five NTDs requiring preventive chemotherapy as a main intervention: lymphatic filariasis, onchocerciasis, schistosomiasis, soil-transmitted helminths and trachoma.

## METHODS
### Study design
A cross-sectional ecological study design was used to examine and compare the NTD, population and internal displacement factors and outcomes across different countries for the 2020–2021 period.

### Neglected tropical disease and population data sources, classifications and stratification
The NTD and population data were obtained from the Expanded Special Project for Elimination of NTDs (ESPEN) data portal, which collates data from national programmes and stakeholders on the various activities related to the implementation of MDA and/or morbidity management and disability.[24] For each country, we summarised (1) the number of NTDs requiring preventive chemotherapy and (2) the estimated country population based on 2020 data.

We then classified the countries endemic with four or five NTDs as '*high NTD number*' to reflect the complexity of managing many diseases and programmes, while the other countries were classified as '*mod(erate)-low NTD number*'. Second, we classified the 10 countries with the highest populations (as an arbitrary threshold) as '*high population*' and others as '*mod-low population*'. Third, we

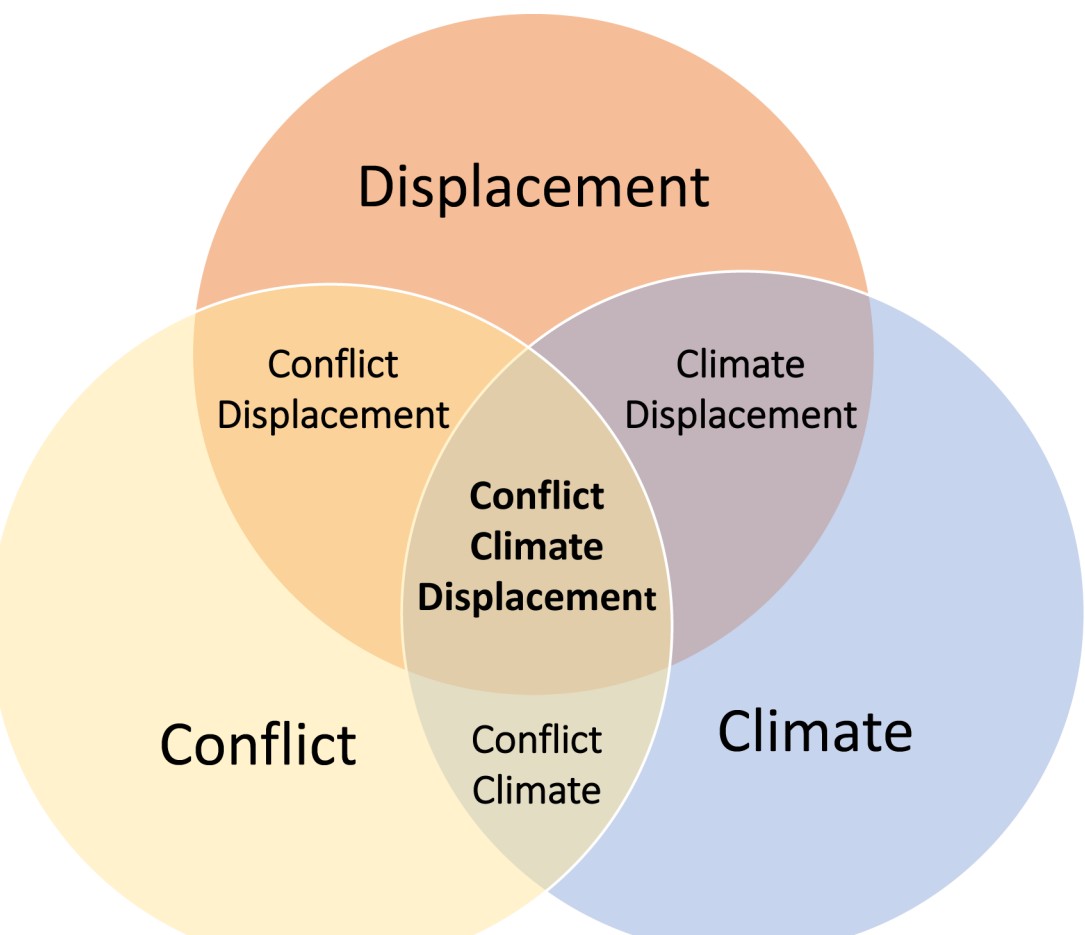

**Figure 1** Intersecting challenges of conflict, climate and related internal displacement.

combined the various classifications to develop four distinct categories, as shown here, and allocated them accordingly to each country.

NTD-population categories:
► High NTD number+high population.
► High NTD number+mod-low population.
► Mod-low NTD number+high population.
► Mod-low NTD number+mod-low population.

### Internal displacement data sources, classifications and stratification

The data on internal displacements were obtained from the IDMC data portal, which monitors and collates country-specific data on internal displacements associated with conflict and generalised violence, and natural disasters (weather-related and geophysical) from a variety of sources.[23]

Internal displacement is referred to as the total number of involuntary or forced movements that have taken place within the borders of a country within a defined period, for example, between 1 January and 31 December 2021, and aims to provide a comprehensive cumulative figure, which may include people being displaced once or several times.

Conflict displacements are referred to as the total number of internal displacements reported as a result of conflict and/or violence events in a reporting year, and natural disaster displacements as the total number of internal displacements reported as a result of natural disasters in a reporting year. For this paper focusing on the concept of climate, the related internal displacements are referred to as natural disasters internal displacements as defined by IDMC and only include countries with reported data for 2021.

For each country, we summarised the number of conflict-related and natural disaster-related internal displacements based on 2021 reported data[23] and calculated crude rates of internal displacement per 100 000 population based on ESPEN population data.[24] Countries with no conflict and climate reported data were not included in the analysis.

We then classified the 10 countries with the highest number and the highest rates of conflict-related or disaster-related internal displacement as '*high conflict*' or '*high disaster*' and others as '*mod-low conflict*' or '*mod-low disaster*'. Second, the conflict and natural disaster classifications were then combined to develop four distinct categories each for numbers and rates as shown here and allocated to each country.

Displacement numbers and rate categories:
► High conflict+high disaster.

► High conflict+mod-low disaster.
► Mod-low conflict+high disaster.
► Mod-low conflict+mod-low disaster.

### Combined categories

All NTD, population and internal displacement categories were then combined to determine the overall patterns, and to help identify countries with high risk or vulnerability. For this paper, we highlighted selected high-risk combinations, which included the countries with a *high NTD number+high population* and a *high NTD number+mod-low population*.

### Drivers of natural disaster-related internal displacement

To better understand the drivers of natural disaster-related internal displacement, we summarised the available IDMC data on hazard types in the countries with the highest natural disaster-related internal displacement numbers and rates in 2021. Hazard types were categorised as climate-related or weather-related (drought, flood, extreme temperature, storm, wet mass movement and wildfire) and geophysical (earthquake, volcanic eruption, dry mass movement). The equivalent data for conflict-related internal displacement in 2021 were not available from this source and therefore not included in this paper.

### Data management and mapping

The NTD, population and internal displacement data and related classifications and risk categories for each country were compiled into a Microsoft Excel file for descriptive analysis and mapping. Maps for all classifications and stratifications were created using mapping software ArcGIS V.10.8.1 (ESRI, Redlands, California, USA) and national administrative boundaries from the ESPEN data portal.[24] First, NTD, population and international displacement individual classifications (eg, *high, mod-low*) were mapped. Second, the different categories (eg, *high+high, high+mod-low, low+low*) of the different factors were mapped to highlight the variety of risk combinations. Finally, the countries with high-risk combinations for all factors (eg, *high+high+high+high*) were mapped to highlight the countries with high risk or vulnerability.

### Data limitations

We acknowledge that the use of open access data sources may have limitations if they are incomplete, and that our focus on a few selected data sources may cause unforeseen biases.

### Patient and public involvement

No patient and/or members of the public were included in this study.

## RESULTS

In total, 45 countries in the WHO African region had data on NTDs and populations (excluding Mauritius, Seychelles), and 32 countries reported data on conflict-related and/or natural disaster-related internal displacements; 16 countries reported both conflict and natural disaster; 15 countries reported natural disaster and one country reported conflict-only internal displacement for 2021.

### Neglected tropical diseases and population at risk

The total population in the 45 countries with data on NTDs was 1.1 billion. Thirty-one countries had a high number of NTDs: 12 countries endemic with 5 NTDs and 19 countries endemic with 4 NTDs (figure 2A). The 10 countries with the highest populations are shown in figure 2B, and of these, 8 had both *high NTD number+high population* (figure 2C) including (in descending order): Nigeria (estimated 197.4 M), Democratic Republic of the Congo (DRC; 105.7 M), Ethiopia (105.7 M), Tanzania (mainland and Zanzibar 55.9 M), Kenya (52.0 M), Uganda (41.0 M), Angola (31.6 M), Mozambique (29.9 M)—totalling >619 M people (table 1).

Twelve countries had *mod-low NTD number+mod-low population* (Botswana, Cape Verde, Comoros, Eritrea, Eswatini, Lesotho, Mauritania, Namibia, Rwanda, Sao Tome and Principe, The Gambia, Togo), while 24 countries had a combination of *high NTD number+mod-low population* and one country (South Africa) had a *mod-low NTD number+high population* (figure 2C).

### Conflict-related and natural disaster-related internal displacement numbers and rates

#### Numbers

The total number of internal displacements in 2021 in the 31 high NTD-endemic countries was 2.2 M. The conflict displacement classifications for each country are shown in figure 2D; the *high conflict* displacement numbers ranged from 0.1 M to 5.1 M across 10 countries: Ethiopia (5.14 M), DRC (2.71 M), Burkina Faso (0.68 M), Central African Republic (CAR; 0.50 M), South Sudan (0.43 M), Mali (0.25 M), Nigeria (0.38 M), Mozambique (0.19 M), Cameroon (0.13 M) and Niger (0.11 M); the *mod-low conflict* numbers across seven countries ranged from 0.94 M to 0.04 M.

The natural disaster displacement classifications are shown in figure 2E; the *high disaster* numbers ranged from 0.24 M to 0.89 M across 10 countries, reported in DRC (0.89 M), South Sudan (0.51 M), Ethiopia (0.24 M), Niger (0.12 M), Burundi (0.09 M), Tanzania (0.05 M), Uganda (0.05 M), Mozambique (0.04 M), Kenya (0.04 M), Nigeria (0.02 M); the *mod-low disaster* numbers in 20 countries ranged from 255 M to 0.02 M.

Six countries had both *high conflict+high disaster*-related internal displacement numbers, including (in descending order) Ethiopia (5.38 M combined), DRC (3.60 M), South Sudan (0.93 M), Nigeria (0.40 M), Mozambique (0.23 M) and Niger (0.23 M) (figure 2F)—totalling >10.8 M conflict and natural disaster combined internal displacements in 2021 (table 1). A summary of the overall internal displacement numbers in the selected high NTD-endemic countries is shown in figure 3A, and highlights

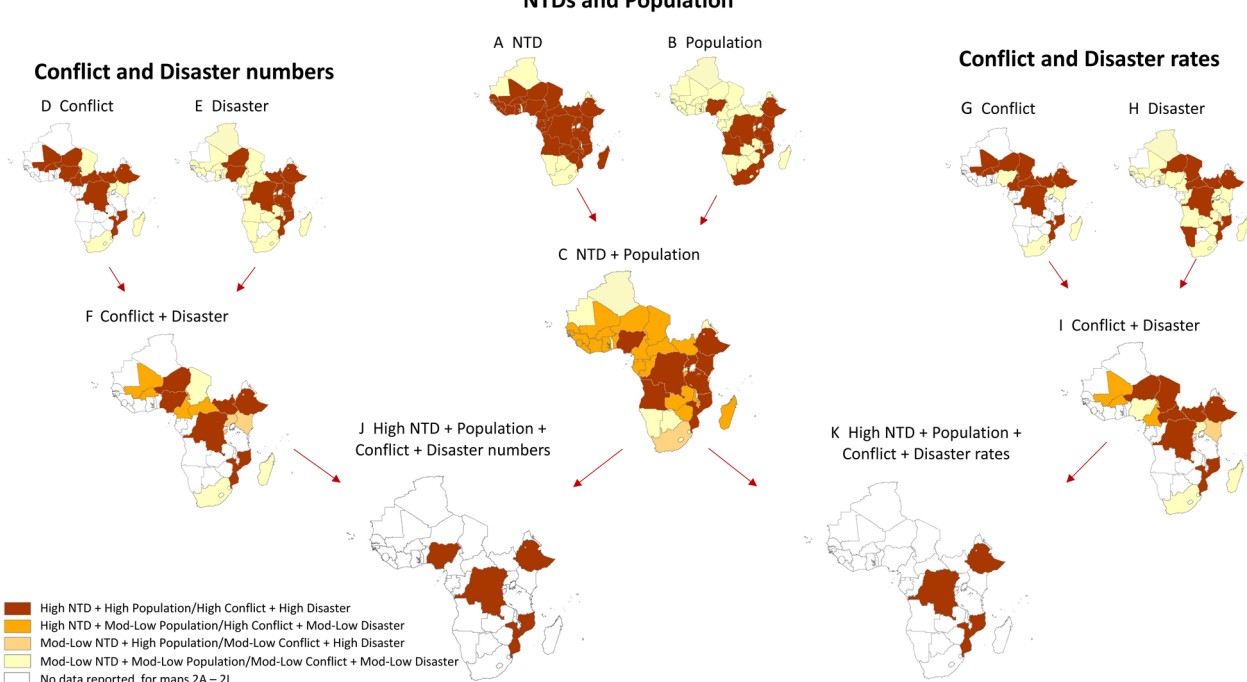

**Figure 2** Summary maps of NTDs, populations and internal displacement numbers, rates classifications and risk combinations.

the descending order and differences between countries with *high populations* and *mod-low populations*.

## Rates

The conflict-related internal displacement classifications are shown in figure 2G; the *high conflict* displacement rates (table 1) ranged from 252.7 to 9020.3 per 100 000 population, reported in CAR (9020.3 per 100 000), Ethiopia (4867.1), South Sudan (3255.6), Burkina Faso (3176.4), DRC (2564.7), Mali (1187.2), Mozambique (625.3), Cameroon (504.0), Niger (475.1), Chad (252.7); the *mod-low conflict* displacement rates ranged from 0.082 to 190.2 per 100 000.

The natural disaster-related internal displacement classifications are shown in figure 2H; the *high disaster* displacement rates (table 1) ranged from 144.1 to 3832.6 per 100 000 population, reported in South Sudan (3832.6 per 100 000), DRC (839.8), Burundi (754.7), Niger (507.6), CAR (429.3), Sao Tome and Principe (237.8), Ethiopia (227.2), Namibia (165.0), Mozambique (145.5), Chad (144.1); the *mod-low disaster* displacement rates ranged from 3.2 to 121.4 per 100 000.

Seven countries had both *high conflict+high disaster* internal displacement rates, including (in descending order) CAR (9449.7 combined), South Sudan (7088.1), Ethiopia (5094.3), DRC (3404.5), Niger (982.7), Mozambique (770.8) and Chad (398.8) (figure 2I). A summary of the overall internal displacement rates per 100 000 in the selected high NTD-endemic countries is shown in figure 3B, and highlights the descending order and differences between countries with *high populations* and *mod-low populations*.

## Risk combinations

Of the countries with *high NTD number+high population*, we found that DRC, Ethiopia, Mozambique and Nigeria had *high conflict+high disaster* displacement numbers (figure 2J), and DRC, Ethiopia and Mozambique had *high conflict+high disaster* displacement rates (figure 2K) in 2021. Collectively, these three countries reported >241 M people at risk of four to five NTDs and 9.2 M conflict and/or natural disaster-related internal displacements in 2021 alone; these may be considered the most at risk or vulnerable countries in the African region. Nigeria was found to have *high conflict+high disaster* displacement numbers, while Tanzania, Kenya and Uganda had among the highest *high disaster* displacement numbers (table 1).

Of the countries with high NTD numbers and displacement, we found South Sudan and Niger to have both *high conflict+high disaster* number and rates, while Burkina Faso, Cameroon, CAR and Mali had both *high conflict* numbers and rates, and Burundi had both *high disaster* displacement numbers and rates in 2021 (table 1).

## Drivers of natural disaster-related internal displacement

The number of reported natural disaster hazard events in 2021 varied across countries, with the highest frequency reported in Burundi (n=80), Uganda (n=60) and South Sudan (n=31), and the highest number of hazard-related internal displacements reported in DRC (0.89 M), South Sudan (0.51 M) and Ethiopia (0.24 M) (table 2). In all countries, weather-related hazards were most common with floods the dominant hazard type, accounting for around half (55.2%) of all natural disaster displacements across 12 countries, except DRC where a volcanic

**Table 1** Summary of populations, and internal displacement data and high-risk combinations in selected countries with high number of NTDs

| Location* | Population† | Internal displacement numbers‡ | | | Internal displacement rates per 100000 population | | | Combined classifications |
|---|---|---|---|---|---|---|---|---|
| Country | Population | Conflict no. | Natural disaster no. | Conflict disaster no. | Conflict rate | Natural disaster rate | Conflict disaster rate | NTD, population conflict, natural disaster |
| **High NTD number+high population** | | | | | | | | |
| DRC | 105.7 M | 2711783 | 888000 | 3.60 M | 2564.7 | 839.8 | 3404.5 | High NTD-population+conflict-disaster |
| Ethiopia | 105.7 M | 5142356 | 240009 | 5.38 M | 4867.1 | 227.2 | 5094.3 | High NTD-population+conflict-disaster |
| Mozambique | 29.9 M | 187057 | 43512 | 0.23 M | 625.3 | 145.5 | 770.8 | High NTD-population+conflict-disaster |
| Nigeria | 197.4 M | 375552 | 24366 | 0.40 M | 190.2 | 12.3 | 202.6 | |
| Tanzania | 55.9 M | 0 | 46707 | 0.05 M | 0 | 85.6 | 85.6 | |
| Kenya | 52.0 M | 4634 | 36442 | 0.04 M | 8.9 | 70.1 | 79.0 | |
| Uganda | 41.0 M | 1278 | 46690 | 0.05 M | 3.1 | 113.8 | 116.9 | |
| Angola | 31.6 M | 0 | 21727 | 0.02 M | 0 | 68.8 | 68.8 | |
| **Summary** | **619 M+** | 8422660 | 1347453 | **9.77 M** | **0–4867.1** | **12.3–839.8** | **68.8–5094.3** | |
| **High NTD number+mod-low population** | | | | | | | | |
| South Sudan | 13.2 M | 429432 | 505544 | 0.93 M | 3255.6 | 3832.6 | 7088.1 | High NTD-conflict-disaster |
| Niger | 23.2 M | 110203 | 117747 | 0.23 M | 475.1 | 507.6 | 982.7 | High NTD-conflict-disaster |
| Burkina Faso | 21.5 M | 682245 | 0 | 0.68 M | 3176.4 | 0 | 3176.4 | High NTD-conflict |
| Cameroon | 26.1 M | 131452 | 1764 | 0.13 M | 504.0 | 6.8 | 510.8 | High NTD-conflict |
| CAR | 5.5 M | 495600 | 23589 | 0.52 M | 9020.3 | 429.3 | 9449.7 | High NTD-conflict |
| Mali | 21.0 M | 249174 | 5994 | 0.26 M | 1187.2 | 28.6 | 1215.7 | High NTD-conflict |
| Burundi | 11.5 M | 94 | 86885 | 0.09 M | 0.8 | 754.7 | 755.5 | High NTD-disaster |
| Chad | 16.8 M | 42355 | 24144 | 0.07 M | 252.7 | 144.1 | 398.8 | |
| **Summary** | **138.8 M** | 2140555 | 765667 | **2.91 M** | **0.8–9020.3** | **0–3832.6** | **398.8–9449.7** | |

Shading represents countries with a classified high NTD number, high population or high conflict and/or high natural disaster-related internal displacement numbers and/or rates.
*All countries have a high number of NTDs.
†ESPEN data 2021.[24]
‡IDMC data 2019–2021.[23]
CAR, Central African Republic; DRC, Democratic Republic of the Congo; ESPEN, Expanded Special Project for Elimination of NTDs; IDMC, Internal Displacement Monitoring Centre; M, million; NTD, neglected tropical diseases.

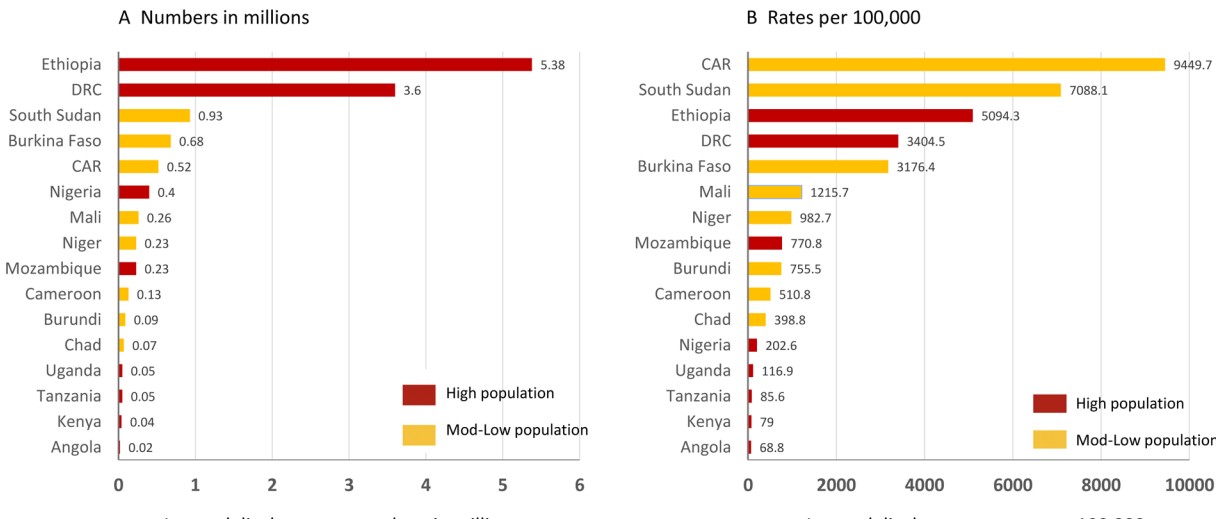

**Figure 3** Summary of overall internal displacement numbers and rates in selected countries with high numbers of NTDs. CAR, Central African Republic; DRC, Democratic Republic of the Congo.

disruption led to 0.6 M displacements, and in Mozambique where a storm led to 0.04 M displacements. In total, 150 flood events across 11 countries caused 0.85

M natural disaster hazard-related internal displacements. Other causes of internal displacement included mass wet or dry movement and wildfires but affected far

**Table 2** Summary of natural disaster-related internal displacements by hazard type in selected countries with high number of NTDs

| Country | Total population | Total no. of natural disaster displacements | Total rate of natural disaster displacements | Total no. of hazard events | No. of natural disaster displacements by hazard type (no. of hazard events) | | | |
|---|---|---|---|---|---|---|---|---|
| | | | | | **Flood** | **Drought** | **Storm** | **Other*** |
| High NTD number, high population, high climate numbers or rates | | | | | | | | |
| DRC | 105.7 M | 0.89 M | 2564.7 | 21 | 0.26 M (n=10) | | 0.002 M (n=8) | 0.6 M (n=3) |
| Ethiopia | 105.7 M | 0.24 M | 839.8 | 7 | 0.19 M (n=4) | 0.05 M (n=1) | | 0.00001 (n=2) |
| Mozambique | 29.9 M | 0.04 M | 145.5 | 2 | | | 0.04 M (n=2) | |
| Nigeria | 197.4 M | 0.02 M | 12.3 | 31 | 0.02 M (n=31) | | | |
| Tanzania | 55.0 M | 0.05 M | 85.6 | 6 | 0.02 M (n=2) | | 0.03 M (n=4) | |
| Kenya | 52.0 M | 0.04 M | 70.1 | 3 | 0.04 M (n=2) | | 0.000 M (n-1) | |
| Uganda | 41.0 M | 0.05 M | 113.8 | 60 | 0.04 M (n=24) | | 0.005 M (n=28) | 0.001 M (n=8) |
| **Summary** | **586.7 M** | **1.33** | **12.3–2564.7** | **130** | **0.57 M (n=73)** | **0.05 M (n=1)** | **0.08 (n=43)** | **0.60 (n=13)** |
| High NTD number, mod-low population, high natural disaster numbers or rates | | | | | | | | |
| South Sudan | 13.2 M | 0.05 M | 3832.6 | 39 | 0.05 M (n=39) | | | |
| Niger | 23.2 M | 0.12 M | 507.6 | 1 | 0.12 M (n=1) | | | |
| CAR | 5.5 M | 0.02 M | 429.3 | 16 | 0.02 M (n=16) | | | |
| Burundi | 11.5 M | 0.26 M | 754.7 | 79 | 0.07 M (n=15) | | 0.013 M (n=63) | 0.0002 (n=1) |
| Chad | 16.8 M | 0.02 M | 144.1 | 6 | 0.02 M (n=6) | | | |
| **Summary** | **70.2 M** | **0.47 M** | **144.1–3832.6** | **141** | **0.28 M (n=77)** | | **0.013M (n=63)** | **0.0002 (n=1)** |

Data source IDMC (numbers rounded).[23]
*Other category included—wet or dry mass movement and/or wildfires.
CAR, Central African Republic; DRC, Democratic Republic of the Congo; IDMC, Internal Displacement Monitoring Centre; M, million; NTD, neglected tropical disease.

A  Proportion of internal displacement numbers

B  Proportion of internal displacement frequency

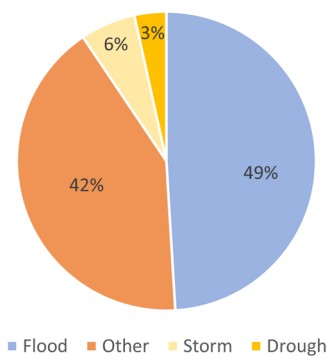

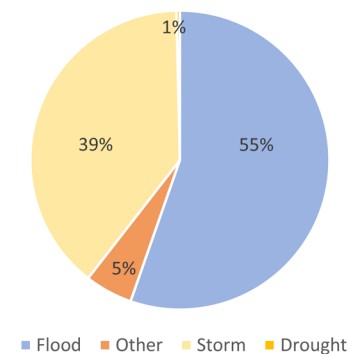

**Figure 4**  Proportion of overall natural disaster-related internal displacement numbers and frequency of events by hazard type.

fewer numbers. A summary of the overall number and frequency of natural disaster-related internal displacements by hazard type in the selected high NTD-endemic countries is shown in figure 4A and B, and highlights the predominance of floods.

## DISCUSSION

This paper highlights that many African countries have high numbers of NTDs, which require a great deal of coordination and logistics in country to implement routine NTD programming.[24] In the eight most populous endemic countries, we quantified that more than half a billion people will need multiple NTD interventions and long-term surveillance activities in the next decade. Routine NTD programming has been and will be negatively impacted by large population displacements, which adds to the existing difficulties of determining accurate population numbers in countries where there is a lack of regular census data.[25 26] The ESPEN data portal provides population estimates[24] and could start to collate data on internal displacements from various sources, including NTD programme managers who are experiencing the challenges first hand and may provide more detailed information and invaluable insights into these problems.

Alarmingly, here we show in just 1 year that Ethiopia, DRC, South Sudan, Nigeria, Mozambique and Niger reported >10 M internal conflict and natural disaster-related internal displacements and have among them the highest conflict and natural disaster rates per population. If current trends continue or increase, these countries are at high risk of not reaching the NTD road map targets and will require significant technical and financial support from stakeholders if global goals are to be met.[1] This is critical as these six countries have among the lowest levels of human development in the world,[27] and may not be able to access and mobilise in-country resources needed to accelerate programmatic action, intensify cross-cutting approaches and change operating models to facilitate enhanced country ownership as set out in the NTD road map 2021–2030.[1]

We limited our analysis to one internal displacement data source but advocate that future research should aim to triangulate information from other data sources to build a comprehensive picture of the situation. We also acknowledge that the use of open-source data may have limitations if they are incomplete or may be updated or changed over time, and that the analyses may only be as good as the available data allow them to be at the time of access. For example, there are frequently issues with the accuracy of the NTD and population data, and additional data on internal displacements for this study may now be available. Therefore, the analyses and outcomes presented in this paper are indicative of the scale of the problem, and numbers should not be taken as being precise.

A further limitation was that we applied simple classifications and risk combinations to the NTD, population and internal displacement figures to allow us to initially focus on countries most affected as a means of understanding the magnitude of these complex challenges. This does not imply that countries with lower numbers or rates of NTDs, population, conflict-related and natural disaster-related internal displacements are less of a priority for NTD programmes, but rather the magnitude is different. All endemic countries in Africa and elsewhere will need support to address these challenges and future work should consider more advanced analysis applied at a subnational level and include additional compounding factors such as health infrastructure, community accessibility and strength of the NTD programme.[15 20]

Furthermore, future work also needs to consider the spatial-temporal patterns and geographical overlap of conflict and natural disaster-driven displacement to identify hotspots, and determine the long-term implications and strategies needed for NTD programme activities. For example, if there is protracted conflict in an area, then programmes can adapt their methods of programmatic delivery, target non-conflict areas or identify methods of treating displaced populations that have sought refuge in secure areas.[17] If flooding is increasingly predictable in an area, then programmes can adapt the timing of

programme activities in that area or identify alternate treatment sites for those displaced by the flooding.

The African continent is predicted to continue experiencing negative effects from climate change due to increasing temperatures, rising sea levels, cumulative failed rainy seasons, drought and shrinking surface water.[8] These events and resulting food and water scarcity can feed into political instability, and exacerbate ongoing tensions thereby creating a vulnerability multiplier for those at risk of conflict, natural disasters and NTDs. Climate change is an escalating security threat,[28–31] and an understanding of the key drivers is critical in Africa and elsewhere. Here, we highlight that most natural disaster hazards were climate-related or weather-related events, especially flooding, which led to nearly 1 M displacements in 11 African countries in 1 year alone. However, the impact of these events on disease transmission dynamics in highly endemic areas, and how they affect programmes, is largely unknown as very little research has been conducted in Africa and elsewhere.[4–6]

Moving forward NTD programmes will need to engage a range of other stakeholders that are working towards addressing and/or mitigating the impact of conflict and/or climate change at an international and national level.[32] This will help to identify country-specific needs and create opportunities for cross-sector partnerships, good coordination and the potential co-production and co-implementation of effective strategies.

## Recommendations

We advocate for a 'call to action' to encourage policy makers, programme managers, researchers, implementing partners and donors to develop, implement and evaluate a set of strategies to better assess NTD burden in areas at risk or experiencing conflict-related and climate-related natural disasters, and deliver interventions that are urgently required. Here, we focus on African countries, but the same methodologies may be applied to different WHO regions to understand the distinct and common challenges. There is a need to build on core principles and adapted survey strategies already being developed and implemented by experienced teams that may address these special populations and problems.[14–16 33 34] It will also be important to learn from countries that have implemented efficient and timely responses during conflicts and disasters to internally displaced populations through cross-sectoral collaboration.

Although national capacity to respond to humanitarian emergencies can be limited, as a first step we suggest an overarching set of recommendations that include the development of preferred practices; data platform; decision support tool, advocacy and communication and implementation research strategies as outlined in box 1.

## CONCLUSION

This paper highlights the urgent need for health policies and guidelines to address the escalating risks of conflict,

> **Box 1  Overarching recommendations to address complex challenges in neglected tropical disease (NTD)-endemic countries**
>
> ⇒ *Preferred practices*: a set of informed preferred practice documents outlining potential step-by-step processes and methods to address singular or multiple intersecting conflict-climate-displacement challenges that impede NTD programmatic activities.
> ⇒ *Data platform*: a central data portal that combines data for each country on: NTD burden, population size, internal and cross-border displacement numbers (by conflict, natural disaster and hazard type categorisation) and administrative boundaries.
> ⇒ *Decision support tool*: a practical online tool that facilitates the national NTD programme teams to input and analysis conflict-climate-displacement data to assess risk and support programmatic decision-making at implementation unit or district level.
> ⇒ *Advocacy and communication*: an advocacy and communication plan with goals to document and inform national and international stakeholders of the key threats and success stories related to the conflict-climate-displacement challenges that NTD programmes confront.
> ⇒ *Implementation research*: a strategic research agenda outlining the range of conflict-climate-displacement challenges, and the potential research questions, methodological approaches and funding opportunities that could be drawn on by NTD programmes and academic partners.

climate change-induced natural disasters and associated internal displacement of populations in NTD-endemic countries in Africa. We have shown that publicly available data sources could potentially be used by national policy makers and NTD programmes as a first step to better understand the intersecting challenges and their level of risk and vulnerability. This would allow them to develop risk assessments and mitigation strategies that may be tailored to the local context and health needs of internally displaced persons, given that the types and burden of NTDs, and the causes of internal displacement, vary within and between countries. There is a need to build new stakeholder collaborations and a collective agenda for these neglected conflict-climate-displacement challenges, so that NTD programmes can meet the 2021–2030 road map targets and 2030 Agenda for Sustainable Development and ensure that no one is left behind.

**Contributors**  LAK-H accepts full responsibility for the work and/or the conduct of the study, had access to the data, and controlled the decision to publish. LAK-H contributed to the conception, methodology, formal analysis, writing—original draft. EMH-E and AS contributed to the interpretation, writing of the recommendations, writing—review and editing. JW and FA contributed to the interpretation, writing—review and editing. All authors approved the final version.

**Funding**  The work was not funded by a specific grant. The University of Liverpool provided funding for the open access publishing cost.

**Competing interests**  None declared.

**Patient and public involvement**  Patients and/or the public were not involved in the design, or conduct, or reporting, or dissemination plans of this research.

**Patient consent for publication**  Not applicable.

**Provenance and peer review**  Not commissioned; externally peer reviewed.

**Data availability statement** All data relevant to the study are included in the article or available in the public, open access repositories.

**Author note** Authors are members of the Neglected Tropical Diseases Non-Governmental Organisation Network (NNN), Conflict and Humanitarian Emergency (C&HE) Cross-Cutting Group-Mapping Task Team. www.ntd-ngonetwork.org/cross-cutting-groups/conflict-and-humanitarian-emergencies.

**ORCID iDs**
Louise A Kelly-Hope http://orcid.org/0000-0002-3330-7629
Emma Michèle Harding-Esch http://orcid.org/0000-0002-1432-8109
Angelia M Sanders http://orcid.org/0000-0002-7284-0303

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
