## [Reviewer comments · BMJ Open]

ARTICLE DETAILS

TITLE (PROVISIONAL)	Conflict-climate-displacement: a cross-sectional ecologic study determining the burden, risk and need for strategies for Neglected Tropical Disease programmes in Africa.
AUTHORS	Kelly-Hope, Louise; Harding-Esch, Emma; Willems, Johan; Ahmed, Fatima; Sanders, Angelia

VERSION 1 – REVIEW

REVIEWER	Sanchez, Ana Lourdes Brock Univ, Health Sciences
REVIEW RETURNED	17-Jan-2023

GENERAL COMMENTS	Thank you for this interesting work underscoring the magnitude of the problem with regards to NTDs programmatic challenges in situations of population displacement. Aside from pointing out one typo in Results line 204 (45 counties), I'd like to offer a couple of comments/suggestions: 1. While it is understandable why you would not want to address the root cause of the problem (i.e., climate change-induced disasters and conflict), I wonder if there could be a way to include some relevant aspects in the discussion as well as in the recommendations. Are there national / international stakeholders working towards reducing conflict or addressing climate change?2. I appreciate the recommendations presented in Box 1. They are well crafted and make sense. However, beyond acquiring data, one has to wonder what is the capacity of these countries (their National NTD programme teams) to maintain the capacity to respond under such circumstances. Can you provide examples of countries that have implemented efficient and timely responses during conflicts and disasters to displaced populations?
---

REVIEWER	Al-Delaimy, Ahmed K Anbar University
REVIEW RETURNED	22-Feb-2023

GENERAL COMMENTS	Thank you for floating this research work. I have few suggestions: - It would be better for the manuscript if you shorten the title of the manuscript.- Large part of the manuscript is a review process from national registration, so do you consider your paper a review paper or retrospective survey study (it should be clarified more in methods section)- The objective within the abstract difficult to follow by the reader! (to determine the burden and risk of conflict and 6 climate related internal displacements and the need for strategies for countries endemic with 7 Neglected Tropical Diseases (NTDs))
---

	 - Too much sub titles - mapping software ArcGIS 10.8.1 needs to have more details on its application. - density of population, political conflicts, flooding and environmental hazards, risk and burden, Neglected diseases. Don't you think it is somehow confusing for readers to follow! I mean on what scientific basis you took these different co-factors collectively? For my opinion it would be more specific and accurate if one concentrate on specific risk factors related to NTDs.  - How come the strength and limitation section was written in the beginning or in the middle of the manuscript ? - large tables can be demonstrated by figures. - Although ethics approval mainly on human participants, still the information brought from national registration instituted at least needs ethical and national permission document. I think the manuscript is somehow confusing with these different categories: population, displacement, combined.
--	---

VERSION 1 – AUTHOR RESPONSE

Reviewer: 1

Dr. Ana Lourdes Sanchez, Brock Univ

Comments to the Author:

Thank you for this interesting work underscoring the magnitude of the problem with regards to NTDs programmatic challenges in situations of population displacement.

Response: Thank you

Aside from pointing out one typo in Results line 204 (45 counties), I'd like to offer a couple of comments/suggestions:

Response: Typo now amended

1. While it is understandable why you would not want to address the root cause of the problem (i.e., climate change-induced disasters and conflict), I wonder if there could be a way to include some relevant aspects in the discussion as well as in the recommendations. Are there national / international stakeholders working towards reducing conflict or addressing climate change?

Response: We appreciate your understanding about why it would be challenging to discuss in-depth the root causes of displacement within this manuscript. Additionally, some of the challenges and programs that address conflict and climate change are usually different than the national programs and organizations that focus on NTDs. We have provided more information on the internal displacement background and definitions to help inform the readers, as well as an additional reference. We hope that this manuscript will increase the discussion between these sectors. In the discussion, we have included reference to

importance to engage with international and national stakeholders that are working toward reducing conflict and/or addressing climate change, which will help to identify country specific needs and create opportunities for cross-sector partnerships. We have added in a number of additional references on NTD groups working in displacement areas in Africa.

2. I appreciate the recommendations presented in Box 1. They are well crafted and make sense. However, beyond acquiring data, one has to wonder what is the capacity of these countries (their National NTD programme teams) to maintain the capacity to respond under such circumstances. Can you provide examples of countries that have implemented efficient and timely responses during conflicts and disasters to displaced populations?

Response: *Thank you for your question. We have updated the language around the recommendations presented in Box 1 to reflect that we understand that national capacity might be limited in these settings. We feel that though it will be challenging to implement some of these recommendations it is worth national programs and supporting partners to try and start to discuss how these may be achieved with the right technical and financial support. We hope that this initial manuscript will lead to future cross-sectoral work. Additionally, we have added some references to stakeholders and examples on cross-sectoral collaboration.*

Reviewer: 2

Dr. Ahmed K Al-Delaimy, Anbar University

Comments to the Author:

Dear Author,

Thank you for floating this research work. I have few suggestions:

- It would be better for the manuscript if you shorten the title of the manuscript.

Response: Thank you for your suggestion to shorten the title. We believe it is informative and reflects the content of the paper well, so would prefer to keep the title the same length.

- Large part of the manuscript is a review process from national registration, so do you consider your paper a review paper or retrospective survey study (it should be clarified more in methods section)

Response: The paper presents an ecological study design as it examines data (variables and outcomes) of an entire population. We have clarified this on page 5 under Figure 1 and at the beginning of the methods.

- The objective within the abstract difficult to follow by the reader! (to determine the burden and risk of conflict and 6 climate related internal displacements and the need for strategies for countries endemic with 7 Neglected Tropical Diseases (NTDs))

Response: Thank you for highlighting this issue. These seem to be typos that may have occurred with conversion during submission that appeared inadvertently. The abstracts should read as follows:

The objective of this study was to determine the burden and risk of conflict- and climate-related internal displacements, and the need for strategies for countries endemic with Neglected Tropical Diseases (NTDs).

- Too much sub titles

Response: *As there are several different aspects to the paper e.g., NTDs, and internal displacements, and risk combinations, we prefer to keep the current list of subtitles as it will help the reader understand the different sections and related sub-sections. However, if the paper is accepted and the proof of the paper indicates that these may be better if they were reduced or altered in any way, then we will liaise with the editorial office and amend this accordingly to ensure that the paper has a logical flow to follow.*

- mapping software ArcGIS 10.8.1 needs to have more details on its application.

Response: *We have added in more information to the methods section to better clarify what mapping process was conducted.*

- density of population, political conflicts, flooding and environmental hazards, risk and burden, Neglected diseases. Don't you think it is somehow confusing for readers to follow! I mean on what scientific basis you took these different co-factors collectively? For my opinion it would be more specific and accurate if one concentrate on specific risk factors related to NTDs.

Response: *Thank you for your concern regarding the multiple factors we examined in this paper. The data on both conflict and climate related internal displacements are available together on the Internal Displacement Monitoring Centre (IDMC) portal and related reports. We believe it was important to examine these data together as they are often considered separately, which could bias or limit the analysis. The linkage between climate change/natural disasters and conflict and vector-borne disease control programmes in Africa is understudied; however, a number of recent papers have raised this as an increasing concern in general as highlighted by our references.*

- How come the strength and limitation section was written in the beginning or in the middle of the manuscript ?

Response: *This is the format of the journal as it showcases each paper's main limitation upfront in the final publication.*

- large tables can be demonstrated by figures.

Response: *We believe the tables included in the paper could be useful to national governments, supporting partners and potential donors and would be hard to understand if*

only provided in figure format. Therefore, we find these tables and current mapping figures complimentary.

- Although ethics approval mainly on human participants, still the information brought from national registration instituted at least needs ethical and national permission document.

Response: *The data for this study were obtained from open-access data sources and any ethical approval would have been the responsibility of those organisations before publication of the data. Using the country-level data for this study required no ethical approval.*

I think the manuscript is somehow confusing with these different categories: population, displacement, combined.

Response: *Thank you for your feedback. We realize that the manuscript covers complex topics such as displacement and disease across multiple countries. For this reason, we created the different categories based on simple classifications (high vs low) to demonstrate the potential overlap of such challenging factors in NTD endemic countries. Our aim was to combine data from different platforms into one location and present this information in a consolidated way. We acknowledged in the discussion that our approach was relatively rudimentary, and more could be done with more datasets over time. Additionally, we hope this manuscript can show readers what is possible and motivate countries and supporting stakeholders to conduct similar activities as a national level. Getting the balance between being too simple and too complex is important, and we believe we have achieved this so it is a contribution to new knowledge!*

VERSION 2 – REVIEW

REVIEWER	Sanchez, Ana Lourdes Brock Univ, Health Sciences
REVIEW RETURNED	23-Mar-2023
GENERAL COMMENTS	Thank you for your clarifications and additions to the manuscript. I find this work highly informative not only for the African continent but elsewhere. With regards to the recommendations in Box 1, you stated "We have updated the language around the

	recommendations presented in Box 1 to reflect that we understand that national capacity might be limited in these settings" However, it seems to me that the text in Box 1 in your revised version is identical to the submission. Perhaps this was an editing oversight?
--	---

REVIEWER	Al-Delaimy, Ahmed K Anbar University
REVIEW RETURNED	07-Apr-2023

GENERAL COMMENTS	Thank you for correcting most of reviewer's comments. Still the reviewer think it is easier that readers and national governmental personnel follow up the statistical results through graphics and figures and not large tables. Also in methodology it needs more clarification about the 'ecological study design' whither it is primary group study was measures or a mixed group measurement study, or is it time trend study?. Again In methods it needs to highlight the ecologic inferences about effects on group rates
---

VERSION 2 – AUTHOR RESPONSE

Reviewer: 1

Dr. Ana Lourdes Sanchez, Brock Univ

Thank you for your clarifications and additions to the manuscript. I find this work highly informative not only for the African continent but elsewhere. With regards to the recommendations in Box 1, you stated "We have updated the language around the recommendations presented in Box 1 to reflect that we understand that national capacity might be limited in these settings" However, it seems to me that the text in Box 1 in your revised version is identical to the submission. Perhaps this was an editing oversight?

***Response:** We had kept the language in Box 1 the same but had modified sentences before Box 1 it to better describe its content. We acknowledge that we may not have written a clear response in our previous correspondence.*

Reviewer: 2

Dr. Ahmed K Al-Delaimy, Anbar University

Thank you for correcting most of reviewer's comments. Still the reviewer think it is easier that readers and national governmental personnel follow up the statistical results through graphics and figures and not large tables. Also in methodology it needs more clarification about the 'ecological study design' whether it is primary group study was measures or a mixed group measurement study, or is it time trend study? Again, in methods it needs to highlight the ecologic inferences about effects on group rates.

***Response:** We have added in two figures to visually highlight key information from Table 1 and 2. We advocate that both Tables are very informative and do not want to omit them – the*

related figures summarise the data in a slight different way to add some further perspective.. We think both the Tables and new Figures are of value. We appreciate the reviewer's efforts to make the information in the tables simplified for readers. However, as current NTD practitioners in many of the countries listed in the tables it is the perspective of the authors of this manuscript that the detailed country level data will be useful for NGO, government ministries, and donors working in this space. It is the detailed country level data regarding displacement and sources of displacement that will make this manuscript a reference piece for future work in this growing thematic space.

Re methodology: We have modified the wording to better inform readers that it was a cross-sectional ecologic study design that examined and compared NTD, population and displacement factors/exposures and outcome across different countries for the 2020-2021 period. This allowed us to compare country data. We do not use a multiple-group design. The cross-sectional wording indicates that it was a snapshot of information for the specified time, we have now added in.